# Renal Function and the Role of the Renin–Angiotensin–Aldosterone System (RAAS) in Normal Pregnancy and Pre-Eclampsia

**DOI:** 10.3390/jcm14030892

**Published:** 2025-01-29

**Authors:** Panagiotis Tsikouras, Konstantinos Nikolettos, Sonia Kotanidou, Nektaria Kritsotaki, Efthymios Oikonomou, Anastasia Bothou, Sotiris Andreou, Theopi Nalmpanti, Kyriaki Chalkia, Vlasios Spanakis, Nikolaos Tsikouras, Christina Pagkaki, George Iatrakis, Christos Damaskos, Nikolaos Garmpis, Nikolaos Machairiotis, Nikolaos Nikolettos

**Affiliations:** 1Department of Obstetrics and Gynecology, Democritus University of Thrace, 68100 Alexandroupolis, Greece; k.nikolettos@yahoo.gr (K.N.); kotanidou.so@gmail.com (S.K.); nektaria.97@hotmail.com (N.K.); eftoikonomou@outlook.com (E.O.); soterisand@hotmail.com (S.A.); theonalmpanti@hotmail.com (T.N.); kikichalkia@yahoo.gr (K.C.); spanakvls@outlook.com.gr (V.S.); nick.tsik.2001@gmail.com (N.T.); christinapagkaki@gmail.com (C.P.); nnikolet@med.duth.gr (N.N.); 2Department of Midwifery, School of Health Sciences, University of West Attica (UNIWA), Egaleo Park Campus-28, Ag. Spyridonos Street, 12243 Egaleo, Greece; natashabothou@windowslive.com (A.B.); giatrakis@uniwa.gr (G.I.); 3Department of Emergency Surgery, Laiko General Hospital, 11527 Athens, Greece; x_damaskos@yahoo.gr; 4N.S. Christeas Laboratory of Experimental Surgery and Surgical Research, Medical School, National and Kapodistrian University of Athens, 11527 Athens, Greece; nikosg22@hotmail.com; 5Department of Surgery, Sotiria General Hospital, 11527 Athens, Greece; 6Department of Obstetrics and Gynecology, Atticon, 12462 Athens, Greece; nikolaosmachairiotis@gmail.com

**Keywords:** RAAS, renin, angiotensin, pre-eclampsia, pregnancy

## Abstract

**Objective:** Pre-eclampsia (PE) is a complex, advancing condition marked either by the recent emergence of hypertension and proteinuria or by the recent onset of hypertension accompanied by notable end-organ impairment, which may occur with or without proteinuria. PE usually appears in the postpartum phase or after the 20th week of gestation, though it might appear sooner in cases of molar pregnancy. **Pathophysiology:** This disorder is a serious multisystem condition during pregnancy that can lead to serious, life-threatening complications for both the mother and the fetus if it is not diagnosed and managed promptly. These changes result from widespread and intense vasoconstriction caused by renin, angiotensin II (Ang II), aldosterone, and catecholamines. The renin–angiotensin–aldosterone system (RAAS) and its sequence of signaling reactions demonstrate its crucial role in regulating blood pressure and electrolyte balance that diverges in PE from that observed in healthy pregnancies. However, it is widely recognized that the above description is incomplete for pre-eclampsia and further relationships should be analyzed. The purpose of this article is to review and synthesize alterations in renal function and the RAAS during normal pregnancy and pre-eclampsia. We aim to provide clinical gynecologists with a comprehensive understanding of how pregnancy-specific adaptations can impact renal function and the RAAS, as well as the deviations observed in pre-eclampsia. **Conclusion:** This information is essential to enhance clinical awareness, improve the accuracy of diagnosis, and support evidence-based decision-making in the management of pregnant patients, especially those complicated by pre-eclampsia.

## 1. Introduction

The kidneys are essential for the long-term control of blood pressure, which occurs through two main mechanisms. First, they control extracellular volume by excreting sodium, which is necessary for pressure diuresis and pressure natriuresis. Secondly, the kidneys secrete active substances that participate in their self-regulation and influence other systems governing blood pressure levels, such as the renin–angiotensin–aldosterone system (RAAS). The precise role of the RAAS in the manifestation of pre-eclampsia (PE) symptoms in the mother remains not fully elucidated. In PE, there seems to be an augmented response to Angiotensin II (Ang II), although this heightened response is not due to a greater presence of Angiotensin II type 1 receptors (AT1R) [1]. Structural and functional alterations in the kidneys and the RAAS can consequently contribute to the development of arterial hypertension [2].

## 2. Pre-Eclampsia

PE is a multisystemic clinical syndrome unique to the human species occurring in 1–5% of all pregnancies. According to new aspects, PE refers to the new onset of hypertension plus significant end-organ dysfunction with or without proteinuria and uteroplacental dysfunction in a previously normotensive patient [3]. Pregnancies diagnosed with pre-eclampsia belong to high-risk pregnancies [4]. Hypertension in pregnancy is characterized by systolic blood pressure ≥ 140 mmHg or diastolic blood pressure ≥ 90 mmHg after 20 weeks of pregnancy in a woman who was normotensive before pregnancy. Hypertension is classified as severe when the systolic blood pressure is ≥160 mmHg and/or the diastolic blood pressure is ≥110 mmHg [5]. Despite the numerous theories proposed over the past few decades, the etiology of PE remains elusive, and investigation into its etiopathogenesis continues to captivate researchers from various fields [6]. Its manifestation is linked to the placenta, and symptoms typically resolve following its removal [7].

The primary prerequisite for the appearance of the syndrome is endothelial dysfunction and heightened sensitivity of blood vessels to the hypertensive effects of various factors, culminating in local ischemia, peripheral vasoconstriction, and cellular dysfunction [7]. In general, the release of placental factors can affect the maternal endothelium, creating the circumstances for endothelial dysfunction and edema, which are typical findings of pre-eclampsia [8]. Furthermore, pre-eclampsia has some similarities to steroid-sensitive nephrotic syndrome, including hypertension, which is related to endothelial dysfunction [9]. Uteroplacental ischemia plays a significant role in PE by triggering the release of renin and activating the Ang II pathways. Simultaneously, it reduces the levels of prostaglandin E (PGE), a potent vasodilator, that normally would offset the hypertensive effects of the RAAS.

## 3. Alterations in Renal Function During Normal Pregnancy

Through pregnancy, there are noticeable alterations in both the structure and function of the kidneys. These modifications are necessary to enable the cardiovascular system to adapt to the increased energy requirements of the fetus and the mother. An increase in cardiac output and a decrease in systemic peripheral resistance are common examples of these adaptations [10,11]. The renal volume increases approximately up to 30% owing to the rise in renal vascular volume, glomerular hypertrophy, and, in particular, the distention of the pelvicalyceal system stimulated by hormonal and mechanical factors. Furthermore, the renal plasma flow (RPF) increases up to 80%, and the GFR rises by 50% [12].

While the increase in GFR during pregnancy is often attributed to increased RPF, studies have shown that by the end of gestation, GFR continues to increase despite a reduction in filtration surface resistance (FSR). This can be explained by the significant reduction in oncotic pressure caused by the hemodilution of plasma proteins secondary to the marked increase in plasma volume [13,14]. The issue responsible for these changes in the function of the glomeruli is considered to be the reductions in resistances in both the afferent and efferent arterioles [15]. Various hormonal factors play key roles in the renal hemodynamic changes observed in pregnancy. Progesterone is implicated in causing renal hemodynamic changes. Conversely, studies in both humans and experimental animals suggest that estrogens have no discernible effect on GFR or RPF [16].

The concentrations of urea and creatinine in the plasma of pregnant women decrease due to elevated GFR. Uric acid levels show a similar decline, which can be ascribed to increased GFR and/or reduced reabsorption in the proximal tubule [12]. Pregnancy-related increases in GFR decreased reabsorption in the proximal tubule, and it is possible that changes in glomerular electrostatic charge are the causes of the relatively higher excretion of urine proteins [17].

Relaxin is a hormone with potent vasodilator properties, secreted by the ovaries during the secretory phase of the menstrual cycle, and its levels surge from the early stages of pregnancy. The most influential stimulant for relaxin secretion during pregnancy is human chorionic gonadotropin (hCG), produced by the placenta [18]. Recent studies have highlighted relaxin’s significance as a crucial mediator of renal vasodilation and hyperfiltration observed during pregnancy. Introducing relaxin to experimental animals results in renal artery impairments akin to those witnessed during pregnancy [19]. Relaxin also has the property of inhibiting leukocyte activation and enhancing Nitric Oxide Synthase (NOS) enzyme expression and Nitric Oxide (NO) synthesis [20]. Through NO mobilization and the activation of Endothelin-1 (ET-1)-specific receptors, relaxin appears to play an important role in the renal vasodilation of normal pregnancy [21].

Plasma concentrations of Ang II and atrial natriuretic peptide (ANP) are elevated in pregnancy [16]. Ang II and ANP exert antagonistic effects in the regulation of plasma volume, blood pressure, and electrolyte balance during pregnancy, even with a 50% increase in GFR [22]. The observation of an augmented total plasma volume during pregnancy, despite elevated levels of ANP, highlights the pivotal role of Ang II in this physiological process [23].

In a typical pregnancy, estrogen and progesterone trigger elevations in the concentrations of angiotensinogen (AGT) and renin, both at tissue and systemic levels, activating the RAAS [10,20]. The increased activities of renin, already observed in the first weeks of pregnancy, and of AGT leads to an increase in the level of Ang II. During pregnancy, resistance to the vasoconstrictive effects of Ang II is observed, resulting in the reduction in blood pressure during gestation [24]. Resistance to Ang II action is associated with the down-regulation of AT1R [19]. Also, as emerged from recent research, it is associated with the formation of specific forms of angiotensin 1–7 (Ang 1–7), with vasodilating properties, as well as a specific enzyme (angiotensin-converting enzyme 2—ACE2). ACE2 resembles an angiotensin-converting enzyme (ACE) but exhibits different biochemical properties characterized by a catalytic action in the production of Ang 1–7, while simultaneously inhibiting the vasoconstrictor action of Ang II [25]. Lastly, aldosterone concentrations in both plasma and urine typically rise in pregnant women [20].

## 4. Alterations in Renal Function During Pre-Eclampsia

The exact mechanisms that lead to changes in the hemodynamic status of the kidneys in PE are not fully clarified. Renal dysfunction is a component of broader endothelial dysfunction, which is believed to contribute to the pathogenesis of the disease by inducing generalized vasospasm and diminished organ perfusion [17].

In PE, GFR and RPF are reduced, resulting in a reduced filtration fraction. The primary mechanisms underlying the reduction in GFR in PE include diminished RPF and a reduction in the glomerular filtration coefficient (Kf), which represents the total capillary surface area available for filtration. The reduction in Kf is due to the structural changes presented by the kidney in pre-eclamptic women, while the drop in RPF is due to the high renal vascular resistance and especially to the increased tone of the afferent arteriole [26]. Although circulating levels of Ang II may remain unchanged during PE, the reduction in uteroplacental perfusion pressure, which leads to placental ischemia, could heighten the sensitivity and resistance of renal arterioles (both efferent and afferent) to Ang II. This increased responsiveness might occur through various mechanisms, including the augmented production of thromboxane, a strong vasoconstrictor. Thromboxane may work in opposition to vasodilators like prostacyclin, contributing to the rise in renal vascular resistance, which further exacerbates hypertension and kidney dysfunction in PE [27].

Albuminuria is a pivotal manifestation of PE syndrome, typically concomitant with hypertension [26]. It manifests as a non-selective type, primarily attributed to alterations in glomerular permeability [26,27]. Changes in the size selectivity of the glomerular barrier and the electrical charge of the filtered proteins coincide with the morphological changes in the glomeruli seen in pre-eclamptic women. The degree of albuminuria usually increases with the degree of PE and kidney damage, and in some circumstances, it can even reach the level of nephrotic syndrome. It is interesting to note that a percentage of women show microalbuminuria up to 3–5 years after the onset of PE, expressing the possible persistence of endothelial dysfunction [28,29].

Renal biopsy in pre-eclamptic pregnant women reveals consistent and almost pathognomonic findings. The set of findings constitute a characteristic picture called “glomerular endothelialization” [30]. The glomeruli exhibit edematous and hypertrophic characteristics, yet they do not display hyperplasia, indicating the absence of an increase in cell count. This hypertrophy is due to swelling mainly of the endothelial but also of the mesangial cells, which occupy the lumen of the capillaries and cause their narrowing or blockage, resulting in the manifestation of glomerular ischemia that appears normal. In some instances, it appears to be diffuse, with its severity correlating with the intensity of PE. Foam cells and fibrin-like material are occasionally detected, while crescent formations are extremely rare. Urinary tubules typically exhibit no distinct alterations [31]. However, in cases of pre-existing chronic hypertension, coexisting nephrosclerosis and glomerular endotheliosis are revealed in the vessels and interstitial space. Electron microscopy reveals increased cytoplasm within the edematous endothelial cells, which exhibit abundant vacuoles and significant changes in lysosomes [32,33]. The basement membrane and podocyte foot processes remain intact. The immunofluorescence technique rarely shows fibrin deposits and immunoglobulins of the IgG and IgM type. Pathological findings typically resolve within 2–3 weeks after delivery. Nonetheless, residual evidence may persist for years following the conclusion of the pre-eclamptic pregnancy [33,34].

PE is also characterized by hyperuricemia, primarily resulting from the increased tubular reabsorption of uric acid. The precise mechanism underlying the aforementioned process remains elusive, yet it seems to be related to the reduction in plasma volume [28,29]. Uric acid values are an important indicator of the course and severity of PE, although according to recent data, they cannot be used to predict maternal and fetal complications [29,30].

## 5. Alterations of the RAAS During Normal Pregnancy

The RAAS is one of the most important homeostatic regulators of blood pressure, as well as fluid and electrolyte balance. The primary components of the system that carry out the enzymatic reactions are renin, AGT, ACE, Ang I, Ang II, AT1R, the angiotensin II type 2 receptor (AT2R), and aldosterone [35]. Besides the traditional localization of the RAAS in the kidney, there is evidence indicating that all elements of the RAAS are synthesized in various tissues, including the brain, heart, ovary, and placenta [36,37]. The uteroplacental RAAS regulates maternal blood flow and contributes to spiral artery remodeling, with its components expressed around these arteries during early pregnancy to support vascular regeneration [1,35]. An imbalance among these hormones can lead to placental ischemia, hypertension, and inadequate remodeling of the spiral arteries [38,39].

The primary biologically active hormone synthesized by the RAAS is Ang II, which binds to AT1R expressed on the surfaces of vascular smooth muscle cells and the adrenal glands, triggering a diverse array of biological responses that impact various systems, including the brain, heart, kidneys, blood vessels, and immune system [40]. AT2R is expressed in the fetal kidney, and its expression decreases during the neonatal period [41]. In adult kidneys, AT2 is found in much lower concentrations than ATI, and its stimulation inhibits cell growth, increases cell apoptosis, causes vasodilation, and is involved in fetal tissue development [42,43]. Healthy pregnant women normally exhibit resistance to the vasomotor effects of Ang II [24]. This resistance is likely attributable to elevated levels of progesterone and prostacyclin during pregnancy [44]. Additionally, at the paracrine level, it plays a role in modulating peripheral perfusion and tissue remodeling, as well as regulating neurotransmitters and ion transport [40].

AGT is a glycoprotein produced by the liver, serving as a precursor of the RAAS [40], for the production of Ang I [35]. AGT mRNA has been identified in various tissues beyond the liver, such as the heart, brain, kidneys, adrenal glands, placenta, and ovaries, as well as the vascular and adipose tissues [40]. In normal pregnancies, the plasma AGT concentration increases during the first 20 weeks, probably due to increased estrogen synthesis, and then remains stable [25,43]. During pregnancy, both low-molecular-weight AGT (LMrA) and high-molecular-weight AGT (HMrA) are present, with HMrA being notably elevated [45].

In reaction to decreased renal perfusion, the paraglomerular apparatus’s juxtaglomerular cells produce renin [1,40]. It functions as an enzyme that catalyzes the proteolytic cleavage of AGT to produce the decapeptide Ang I, the initial step in the activation of the RAAS [40]. By eight weeks of gestation, plasma renin activity increases approximately fourfold, further rising to a sevenfold increase by term, reflecting the dynamic adaptations of the RAAS during pregnancy [17,18].

Last but not least, the adrenal cortex’s zona glomerulosa produces the steroid hormone aldosterone, which has an impact on the kidneys by reducing the quantity of sodium eliminated in the urine [46]. The main hormones that stimulate the production of aldosterone are Ang II, the adrenocorticotropic hormone (ACTH), and the concentration of extracellular potassium levels [40,47]. Plasma aldosterone concentrations increase by approximately 3- to 8-fold early during normal pregnancy compared to nonpregnant levels, reaching a plateau in the third trimester. This elevation reflects the physiological adaptations of the RAAS to support increased fluid retention and vascular demands during gestation [20].

## 6. Alterations of the RAAS in Pre-Eclampsia

The RAAS plays an important role in arterial hypertension and, more broadly, in the pathogenesis of PE [32,33,34]. Women with PE demonstrate lower levels of renin, aldosterone, Ang I, and Ang II compared to healthy pregnancies, but the low-renin state raises concerns about inadequate RAAS regulation [46]. Women with PE also exhibit increased sensitivity to Ang II, which is particularly evident in both the adrenal cortex and the vasculature. The heightened responsiveness is associated with the altered regulation of the RAAS observed in PE, contributing to elevated blood pressure and vascular dysfunction, characteristic of the condition [1,39,48]. Furthermore, Ang 1–7, a key counter-regulator of Ang II, plays a protective role by promoting vasodilation and counteracting the vasoconstrictive effects of Ang II. In PE, Ang 1–7 levels are often dysregulated, contributing to the vasoconstrictive and hypertensive features of the disease. Experimental studies have demonstrated that impaired placental perfusion in PE results in increased local Ang II production, although Ang II levels are typically reduced systemically in PE [49].

In contrast to a normal pregnancy, studies indicate that 47% of women with pregnancy-induced hypertension (PIH) exhibit a significantly increased plasma HMrA/LMrA AGT ratio [48]. Yilmaz et al. demonstrated a study examining urinary AGT levels in pre-eclamptic pregnancies. Women at approximately 35 weeks of gestational age in normal pregnancies presented significantly higher urinary AGT levels compared to those in pre-eclamptic pregnancies and nonpregnant women [50]. Reduced urinary levels of AGT in patients with PE suggest the suppression of the local renal RAAS. This suppression is associated with key clinical features of PE, including hypertension and proteinuria, indicating a disruption in renal regulatory mechanisms [51].

Aldosterone, in addition to Ang II, plays a crucial role in regulating sodium and fluid retention during pregnancy [46]. Aldosterone levels are typically elevated to meet the increased circulatory demands, ensuring proper hemodynamic regulation [19]. In PE, however, this balance is disrupted, contributing to the pathological changes observed in renal function. In cases of increased secretion of aldosterone, the renin activity is being suppressed, leading to PE [1,17,39]. The concentration of aldosterone in pre-eclamptic pregnancies is significantly reduced, which may be related to the reduction in plasma volume presented [46,52]. Low-renin hypertension may be linked to cases of primary aldosteronism [46]. It is, therefore, speculated that the reduction in intravascular plasma volume leads to reduced placental perfusion with subsequent ischemia and a systemic hypertensive response. Reduced aldosterone availability in PE has been associated with loss-of-function polymorphisms of the CYP11B2 gene [53].

The regulation of blood pressure and vascular remodeling is mediated by Ang II binding to the AT1R. The AT1R gene is located on chromosome 3q24, and polymorphisms in this receptor have been associated with increased susceptibility to PE, as they enhance vascular sensitivity to Ang II [54]. AT1R forms a heterodimer with bradykinin (B2), and these AT1/B2 heterodimers are resistant to the inactivation of reactive oxygen species (ROS), increase the production of Nitric Oxide (NO) and are hypersensitive to Ang II [55].

On the contrary, AT2R exhibits opposing effects by promoting vasodilation and inhibiting cell growth. Experimental studies suggest that the activation of AT2R may induce atherogenic effects, including the apoptosis of smooth muscle cells, activation of nuclear factor κB signaling, and upregulation of chemokines [56,57]. Animal studies have shown that the lack of AT2R is associated with the prolonged presence of inflammation [54]. The conventional perspective regarding the significance of AT2R is that they counteract the stimulatory impact of AT1R on tissue hypertrophy and hyperplasia [43]. However, it has been established that the dynamic balance of the two receptors and the combination of the triggering action of the AT1R signaling pathway and the inhibitory action of the AT2R signaling pathway is what leads to the functionality of Ang II in blood pressure, as well as the cardiac hypertrophy and hypertrophy of the arteries and smooth muscle [43,58].

In 2024, Kokori et al. reported the presence of AT1R autoantibodies in pre-eclamptic women (AT1-AA) [59]. Subsequently, many studies have been performed confirming the increased concentration of AT1-AA in pre-eclamptic women compared to normal pregnant women, which could cause these kinds of symptoms in pre-eclamptic women, indicating its important role in the occurrence of this pathology. Such autoantibodies could contribute to endothelial dysfunction and vascular damage. In addition, autoantibodies may induce the production of PE-related products such as ROS, NADPH oxidase, and intracellular calcium nuclear factor-κB; the secretion of sFLT1 and endothelin-1; and increased factor TF in vascular smooth muscle cells [43,58]. Also, studies have shown that they increase levels of the plasminogen activator inhibitor (PAI-1) in trophoblast cells, leading to reduced fibrinolytic activity. The elasticity of trophoblasts decreases during the action of AT1-AA [59].

Various factors have been suggested as potential triggers to produce autoantibodies. However, the exact cause remains unidentified. These factors encompass immune responses, genetic predisposition, and environmental influences [59]. AT1-AAs are produced by mature B-lymphocytes (CD19+ and CD5+) and appear only in pre-eclamptic placentas and not in non-pre-eclamptic ones, and they can be detected even 18 months after delivery [60]. The importance of recognizing the action of AT1-AA lies in the fact that if they are detected early in pregnancy and given the corresponding drugs that will block the activation of the AT1R, then the pre-eclamptic symptoms can be reduced or disappear, reducing the risk for mothers and newborns [61].

Table 1 and Figure 1 summarize the aforementioned changes.

It is crucial to take into account other potential pathogenic processes, especially those related to hormone regulation and the mineralocorticoid receptor (MR). The MR’s involvement in the pathophysiology of PE has been emphasized by recent research. The MR is primarily involved in regulating sodium balance, and its activation by aldosterone contributes to increased blood pressure through sodium and water retention. The dysregulation of MR activity in PE, particularly in renal tissues, exacerbates the hypertensive state. Variants in the MR gene have been associated with increased susceptibility to PE [47,62]. These genetic variants may lead to altered receptor sensitivity to aldosterone, thereby enhancing sodium retention and contributing to hypertension in pre-eclamptic patients. Furthermore, whereas high levels of progesterone usually counteract the effects of MR, in PE, their decrease may be caused by circulating aldosterone-induced organ damage, as well as placental injury. The pathophysiological alterations associated with PE, such as hypertension and renal failure, are further exacerbated by this hormonal imbalance [46], Figure 2.

## 7. Discussion

It is evident that the circulatory RAAS plays an important role in both healthy pregnancy and the pathogenesis of PE by regulating the production of angiogenic factors through AT1R. It appears that the angiogenic system and the circulatory RAAS are not separate entities but can synergistically cooperate to regulate the expression of angiogenic genes and the production of growth factors [63,64]. Multiple studies have described alterations in the RAAS that are associated with pregnancies complicated by PE, which are summed on Table 2.

Aldosterone levels increase during pregnancy and are relatively low in pregnancies affected by PE, which is consistent with alterations in renin and Ang II levels. Low aldosterone levels may be responsible for reduced volume growth and resultant poor placental appearance in PE, making it an important target gene. In cultures of human trophoblasts, cell proliferation increased after the stimulation of the aldosterone gene, demonstrating the importance of its lack of action in the disorder [61,63].

Also, angiotensinogen, renin, ACE, and the AT1R are expressed in the spiral arteries of the endometrium [55]. Therefore, in pregnant women, the endometrium and fetal placenta contain all the necessary components for a functional system RAAS. In these women, levels of renin, Ang I, and aldosterone are lower, unlike ACE levels, which remain relatively constant [43,65]. Women with PE demonstrate heightened adrenocortical and vascular sensitivity to Ang II, whereas non-pre-eclamptic pregnant women typically exhibit reduced vascular sensitivity [35]. In non-pre-eclamptic pregnancy, AT1Rs are rendered inactive by ROS, resulting in reduced sensitivity to binding with Ang II, as mentioned above [55]. Moreover, in comparison to non-pre-eclamptic pregnancies, there is a relatively elevated level of aldosterone, despite the lower level of renin in pregnancies complicated by PE [1,17].

In pre-eclampsia, endothelial dysfunction plays a key role in both proteinuria and RAAS activation, further exacerbating vascular damage. This dysfunction, along with glycocalyx degradation, is central to the pathophysiology of pre-eclampsia [8] and is similar to the mechanisms observed in Minimal Change Disease (MCD) [66]. Interestingly, some patients with steroid-sensitive nephrotic syndrome prior to conception developed pre-eclampsia [9].

Estrogen stimulates the synthesis of AGT, resulting in an increase in circulating Ang II levels during the first 20 weeks of pregnancy. In PE, there appears to be a relative increase in HMrA [67]. Polymorphisms of the Ang II gene that increase plasma AGT levels have also been associated with PE, as demonstrated by Shahvaisizadeh et al. in 2012, with the mechanism involving the reduction in renin, which is a consequence of these increased Ang II levels [66].

Reduced vascular responsiveness to Ang II is linked to normal pregnancy, while PE is linked to increased sensitivity to Ang II, which may occur before the disease’s clinical signs and symptoms. Ang 1–7 is a key component of the RAAS and exhibits antagonistic actions of Ang II, acting as a regulator of vascular tone and releasing NO and prostaglandins [43,55].

Ang 1–7 is generated from Ang II by ACE-2. Plasma angiotensin 1–7 is increased in normal pregnancy and decreased in PE [56,66]. Serum immunoglobulins from women with PE stimulate the AT-1 receptor, but those from healthy pregnant women show no effect.

Genetic linkage studies have revealed the presence of polymorphisms in the genes encoding key components of the RAAS and account for variation in the activity of this system. The precise interactions among all components of the RAAS during both normal and abnormal pregnancies remain incompletely understood. While there is a general upregulation of the RAAS in normal pregnancy, this delicate balance is disrupted in PE. The central role of the placenta in PE is undisputed, as incomplete placentation is implicated in the entire pathogenesis of this disorder [68].

In studies of polymorphisms and their association with diseases, it is emphasized that an allele that promotes the occurrence of PE in this particular case acts in one of the following three ways: Firstly, by causing the dysfunction of physiological mechanisms; secondly, by hindering the actions of other factors; and thirdly, by interacting with another genetic marker. Hence, research is anticipated to be conducted to investigate how the C5312T rs12750834 allele contributes to the development of PE.

Pre-eclampsia is associated with dysregulation of the renin–angiotensin–aldosterone system (RAAS), contributing to hypertension and endothelial dysfunction, as previously discussed. However, the use of conventional RAAS inhibitors during pregnancy is generally contraindicated due to potential adverse fetal effects. As a result, alternative therapeutic strategies are currently under investigation.

One promising approach involves the blockade of Angiotensin II type 1 receptor antagonistic autoantibodies (AT1-AA). Elevated levels of AT1-AAs have been observed in pre-eclamptic patients, and these autoantibodies are known to contribute to disease pathology. Experimental studies using animal models have demonstrated that AT1-AA blockade can improve vascular function, highlighting its potential as a therapeutic avenue [69,70].

Another strategy focuses on targeting the underlying disease mechanisms, including oxidative stress, antiangiogenic factors, and the angiotensin pathway. Certain novel therapeutic agents, such as bioactive peptides derived from food proteins, have shown ACE inhibitory activity in vitro, suggesting their potential as alternative therapeutic options. However, these findings remain preliminary, and further clinical trials are required to evaluate their safety and effectiveness in pregnant populations [71].

In summary, while traditional RAAS inhibitors remain unsuitable for use during pregnancy due to safety concerns, alternative therapeutic strategies targeting components of the RAAS or related pathways are actively being explored. Further research is essential to validate these approaches and ensure their efficacy and safety in managing pre-eclampsia. Considering that inhibitors of the renin–angiotensin–aldosterone system, particularly angiotensin-converting enzyme inhibitors, are commonly used in the treatment of hypertension in the general population but not during pregnancy, treatment strategies related to this system are not included in this study [72].

## 8. Conclusions

Stable or reduced RAAS activity in the second half of pregnancy is a characteristic of hypertensive pregnancies. Unaltered Ang peptide levels and decreased aldosterone secretion accompany this altered regulation, suggesting disturbed physiological processes are necessary for preserving vascular tone and fluid balance during pregnancy. Creating a biomarker for early detection or creating specialized treatments for the illness would be the ultimate objectives of related research.

To better understand the pathophysiological mechanisms and guide the best treatment approaches for PE and hypertension in general, more research into certain polymorphisms and renin can help to determine the sensitivity of the RAAS to Ang II. Future research could be directed toward the study of the genetic and hereditary origin of the polymorphism, as well as the inhibition of renin expression through the mechanism that causes this polymorphism. Also, studies worldwide have shown that biomarkers can significantly help with the diagnosis, prevention, and treatment of PE and are a very important and promising aspect of the treatment of this disorder.

This article aims to examine and integrate changes in renal function and the RAAS during normal pregnancy and pre-eclampsia. Our aim is to equip clinical gynecologists with a thorough understanding of how pregnancy-related adaptations affect renal function and the RAAS, as well as the abnormalities that occur in pre-eclampsia. By providing this insight, we hope to enhance clinical awareness, refine diagnostic accuracy, and promote evidence-based approaches for managing pregnant patients, particularly those facing the challenges of pre-eclampsia.

## Figures and Tables

**Figure 1 jcm-14-00892-f001:**
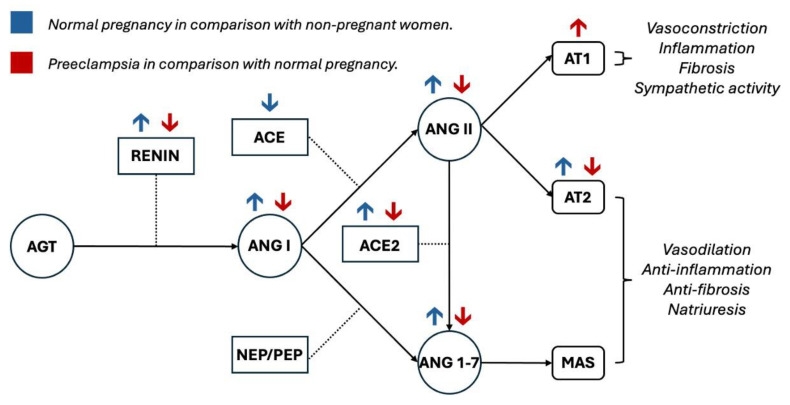
Renin–angiotensin–aldosterone system alterations in non-pre-eclamptic pregnancy and pre-eclampsia. AGT: angiotensinogen; ACE: angiotensin-converting enzyme; ACE 2: angiotensin-converting enzyme 2; ANG I: angiotensin I; ANG II: angiotensin II; AT1: angiotensin type 1 receptor; AT2: angiotensin type 2 receptor; ANG 1–7: angiotensin 1–7; MAS: mas receptor.

**Figure 2 jcm-14-00892-f002:**
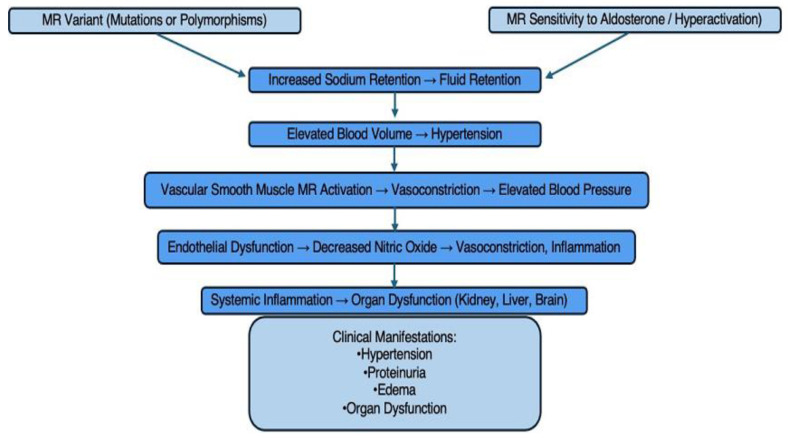
The mineralocorticoid receptor (MR) variants’ involvement in the pathophysiology of pre-eclampsia.

**Table 1 jcm-14-00892-t001:** Renin–angiotensin–aldosterone system alterations in pre-eclampsia.

Reduced Levels of Renin
Reduced levels of aldosterone
Reduced levels of angiotensin I and II
Vasoconstriction
Suppression of renal secretion
Increase in sodium and water retention
Adrenal aldosterone production

**Table 2 jcm-14-00892-t002:** Renin–angiotensin–aldosterone system alterations in pre-eclampsia.

Study	Year	Key Findings	Conclusion
Seki H [61]	2014	Plasma renin and Ang II lower in PE despite hypertension; pressor response to Ang II predicts PE	The circulatory RAAS appears suppressed in PE; the tissue RAAS offers new insights into its pathogenesis
LaMarca et al. [42]	2011	High AT1R expression and AT1-AA in decidua in PE; AT1-AA combination crosses the ureteroplacental barrier	These components may contribute to PE pathophysiology
Yart et al. [64]	2022	Increased renin expression and activation of the uteroplacental RAAS in PE; novel Ang II-related mechanisms	Ang II-mediated mechanisms explain primary features of PE
LaMarca et al. [42]	2012	Immune mechanisms like AT1-AA play a role in PE, AT1-AA induce signaling in vascular cells and trophoblasts, leading to tissue factor production and reactive oxygen species generation.	Important role for AT1-AA stimulated in response to placental ischemia that caused hypertension during pregnancy.
Zhang H et al. [43]	2017	Maternal and fetal polymorphisms in ACE I/D, ACE G2350A, AGT M235T, and AT1R A1166C associated with PE in Han Chinese women	Fetal ACE I/D, ACE G2350A, AGT M235T, and AT1R A1166C polymorphisms significantly influence PE development
ProcopciucLM et al. [53]	2019	Specific RAAS gene polymorphisms independently associated with the risk of early- and late-onset PE	These findings highlight the role of RAAS genetic variations in PE susceptibility
Gintoni I et al. [65]	2021	ACE I/D affects ACE expression and Ang II levels, impacting arterial pressure and fibrinolytic activity	ACE I/D polymorphism linked to pregnancy complications such as PE and aid in predicting pregnancy complications
Shahvaisizadeh F et al. [66]	2014	Some polymorphisms of AGT (217 G→A and T704C) tend to be higher in early-onset PE compared with that in patients with late-onset PE	AGT variants -217 G→A and T704C may synergistically increase risk of severe PE

ANG II: Angiotensin II; PE: Pre-eclampsia; RAAS: Renin–angiotensin–aldosterone system; AT1: Angiotensin type 1 receptors; AT1-AA: Angiotensin II type 1 receptors autoantibodies; ACE: Angiotensin-converting enzyme; AGT: Angiotensinogen.

## Data Availability

Not applicable.

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
