# Peer review of "Renal Function and the Role of the Renin–Angiotensin–Aldosterone System (RAAS) in Normal Pregnancy and Pre-Eclampsia"

_jcm, 2025, doi:10.3390/jcm14030892_

Round 1
Reviewer 1 Report
Comments and Suggestions for Authors
The present review examines renal and RAAS changes in normal pregnancy and preeclampsia to support improved clinical diagnosis and management. The topic is of great interest the the review is overall well organized and written. The description of the pathophysiological mechanisms in preeclampsia should be expanded, emphasizing how endothelial dysfunction contributes not only to proteinuria but also to RAAS activation. Please find below specific comments:
- Consider rephrasing the title for conciseness such as: "Renal Function and the Role of the Renin-Angiotensin-Aldosterone System (RAAS) in Normal Pregnancy and Preeclampsia."
- In preeclampsia, endothelial dysfunction plays a key role in both proteinuria and RAAS activation, further exacerbating vascular damage. This dysfunction, along with glycocalyx degradation, is central to the pathophysiology of preeclampsia (PMID: 37857160) and is similar to the mechanisms observed in Minimal Change Disease (MCD) (PMID: 35497798). Interestingly, some patients with steroid-sensitive nephrotic syndrome prior to conception developed preeclampsia (PMID: 28509244).
- Figure 1 is useful for understanding the alterations in serum levels of various RAAS factors; however, it would be valuable to include an additional figure illustrating the pathophysiological mechanisms underlying these variations.
Author Response
Dear Reviewer,
We would like to thank you for your valuable feedback and constructive suggestions, which have helped us improve the clarity and impact of our manuscript.
In response to your comments:
- We have rephrased the title as per your suggestion to better reflect the focus and objectives of our work.
- A dedicated paragraph discussing the potential treatment has been added.
- To further enhance the understanding of RAAS dysregulation in preeclampsia, we have included a new figure. This illustration outlines the pathophysiological mechanisms and their interconnections, providing a clearer visualization of the alterations in RAAS factors and their clinical implications.
We believe these revisions address your concerns and contribute to a more comprehensive and insightful presentation of our findings.
Thank you again for your thoughtful comments and continued support. We look forward to your feedback on the revised manuscript.
Sincerely,
Prof. Tsikouras Panagiotis

Reviewer 2 Report
Comments and Suggestions for Authors
To the Authors,
Thank you for the opportunity to review your manuscript. This review provides detailed and comprehensive information on the involvement of the renin-angiotensin system in hypertensive disorders of pregnancy. The topic addresses a significant clinical issue and will be valuable for readers interested in this area.
However, to further enhance the quality of this review, I would like to offer the following suggestion:
While the manuscript includes a detailed discussion of the pathophysiology, it provides limited coverage of treatment strategies. Including information on current treatments and potential future therapeutic options suggested by the findings of this review would significantly increase its clinical relevance and impact.
I hope this suggestion helps improve your manuscript and makes it even more beneficial for the readers.
Author Response
Dear Reviewer,
Thank you for your helpful feedback and suggestions, which have improved the clarity and quality of our manuscript.
In response to your comments, we have added a paragraph discussing therapeutic strategies focused on RAAS. This section highlights current and emerging approaches related to the renin-angiotensin-aldosterone system and their potential role in managing preeclampsia.
We believe these updates address your concerns and enhance the manuscript's overall impact.
Thank you again for your support. We look forward to your feedback on the revised version.
Sincerely,
Prof. Tsikouras Panagiotis

Round 2
Reviewer 1 Report
Comments and Suggestions for Authors
The Authors have appropriately addressed my concerns and I do not have any further comment.